# Proteomic Analysis of Mechanical Injury Effects in Papaya Fruit at Two Maturity Stages

**DOI:** 10.3390/proteomes13030044

**Published:** 2025-09-18

**Authors:** Francisco Antonio Reyes-Soria, Eliel Ruiz-May, Enrique Castaño, Miguel Ángel Herrera-Alamillo, José Miguel Elizalde-Contreras, Samuel David Gamboa-Tuz, Lidia F. E. Huerta-Nuñez, Jesús Alejandro Zamora-Briseño, Luis Carlos Rodríguez-Zapata

**Affiliations:** 1Unidad de Biotecnología, Centro de Investigación Científica de Yucatán, Calle 43, No. 130, Chuburná de Hidalgo, Merida CP 97205, Mexico; antonio.ry.16@hotmail.com (F.A.R.-S.); mianheal@cicy.mx (M.Á.H.-A.); sd.gamboa.t@gmail.com (S.D.G.-T.); 2Laboratorio de Farmacología, Escuela Militar de Graduados de Sanidad, Universidad del Ejército y Fuerza Aérea, Ciudad de Mexico CP 11200, Mexico; pelecas40@gmail.com (E.R.-M.); lidi7713@gmail.com (L.F.E.H.-N.); 3Unidad de Bioquímica y Biología Molecular de Plantas, Centro de Investigación Científica de Yucatán, Calle 43, No. 130, Chuburná de Hidalgo, Merida CP 97205, Mexico; enriquec@cicy.mx; 4Red de Estudios Moleculares Avanzados, Campus III, Instituto de Ecología, Carretera Antigua a Coatepec 351, El Haya, Xalapa, Veracruz CP 91070, Mexico; jose.elizalde@inecol.mx

**Keywords:** papaya fruit, post harvesting, proteomics, mechanical damage, bioinformatics approach

## Abstract

Background: Mechanical damage to fruit during harvesting is nearly inevitable, with certain species, such as papaya, being particularly prone to spoilage. Postharvest handling can induce mechanical injuries that impair ripening and reduce shelf life, leading to significant economic losses. Although several studies have shed light on the molecular bases of mechanical damage, other aspects remain to be described (plant hormone inter-talk, physiological changes, and regulatory networks). Methods: In this study, we investigated proteomic changes in papaya fruit at two distinct ripening stages following mechanical damage. A total of 3230 proteins were identified, representing the most comprehensive proteomic analysis of papaya to date and the first assessment of proteins regulated by mechanical stress. Results: Proteins involved in ethylene biosynthesis were up-regulated on Day 2 but down-regulated on Day 12, with a similar trend observed for proteins in the abscisic acid synthesis pathway. Enzymes associated with photosynthesis, carbon fixation, primary metabolism, and carotenoid synthesis were down-regulated at both stages. In contrast, those related to plasmodesmata, calcium signaling, kinases, pathogenesis, cell wall remodeling, and proteases were up-regulated. Conclusions: These findings are thoroughly discussed, and a general model of the events triggered by mechanical impact in papaya is proposed. Our results provide a comprehensive framework for understanding papaya’s response to mechanical damage.

## 1. Introduction

Global food security and economic stability are significantly affected by postharvest losses, a substantial portion of which is attributable to mechanical damage during fruit harvesting and handling [1,2]. This issue is particularly critical for highly perishable fruits like papaya (*Carica papaya* L.), where mechanical injuries disrupt ripening, drastically reduce shelf life, and lead to major economic losses across the supply chain [1,2]. Despite the widespread occurrence and economic impact of postharvest mechanical damage, the intricate molecular mechanisms underlying fruit mechanical damage remain poorly understood [3,4].

As a climacteric fruit, papaya exhibits a sharp and transient rise in respiration during ripening, accompanied by increased ethylene production and sensitivity [4]. This physiological shift activates key biochemical pathways essential for fruit ripening [5], resulting in rapid pulp softening and an inherently short postharvest shelf life (PSL). Additionally, the fruit’s large size and weight make it highly susceptible to impact and compression during handling, transportation, and storage, further increasing its vulnerability to quality loss and pathogen infections. Consequently, papaya is considered a highly perishable commodity, with significant economic losses frequently occurring along its value chain [6,7,8].

While most research on papaya ripening has focused on the “Solo” variety, Santamaría-Basulto et al. [9] established specific ripening indices for the commercially important “Maradol” papaya, defining six distinct ripening stages that serve as quality benchmarks. In this study, we examined two of these stages: an early and a late ripening stage. Moreover, most of the current studies rely on single timepoint proteomic sequences, not necessarily comparing the initial and final state of ripening. Therefore, the present work establishes an appropriate focus to fill the gap regarding the ripening process with a proteomic approach. To date, little is known about how mechanical damage accelerates ripening, or the specific biochemical changes that occur in papayas at defined stages following injury.

The primary effects of mechanical damage in papayas include discoloration, accelerated softening, staining, and bruising [10]. Bruising, a common form of mechanical injury, occurs when excessive external force—whether from impact, compression, or vibration—causes subcutaneous tissue damage without necessarily breaking the skin [11]. These disruptions begin at the microscale, altering cellular structure and integrity, with macroscale effects subsequently reflected by observable morphological changes [12]. Thus, mechanical damage not only causes physical deformities and tissue rupture, but also triggers cellular breakage and complex physiological responses linked to poorly understood metabolic shifts [13].

This knowledge gap is critical, as a deeper understanding of these metabolic changes could inform effective strategies to mitigate postharvest losses [14]. Such insights are essential for developing science-based approaches to reduce losses in production and export supply chain systems [15,16].

Proteomics has become an indispensable tool in plant science, facilitating the comprehensive characterization of proteins involved in metabolic processes that influence postharvest shelf life (PSL) across various fruit species [17,18,19,20,21,22,23]. However, despite its significant potential, proteomic data on papaya ripening—particularly studies focusing on mechanical damage—remain limited [24,25,26,27]. Given that mechanical damage accelerates natural ripening, the scarcity of molecular and proteomic research on bruised papayas represents a critical knowledge gap. Addressing this gap is essential not only for papaya but also for other understudied tropical fruits, where protein-level responses to mechanical damage are still poorly understood [7,28,29]. Moreover, database integrity and precise protein annotation are vital for the reliable detection of proteoforms, defined as all the possible structures which proteins from a single gene can adopt at specific conditions, which play a direct role in regulating cellular mechanisms and abiotic stress responses. Although recent advancements in mass spectrometry (MS) and top-down proteomics have improved proteoform characterization, challenges remain in their comprehensive detection, quantification, and functional annotation [30].

Against this backdrop, our study employs quantitative proteomics to identify molecular changes in bruised “Maradol” papayas at two ripening stages. We hypothesize that mechanical damage induces specific metabolic responses, altering the abundance of proteins linked to oxidative stress, protein degradation, and the phenylpropanoid pathway.

## 2. Materials and Methods

### 2.1. Plant Material

Mature green papaya fruits (*Carica papaya* L. cv. Tainung) were sourced from commercial plantations operated by PAMASUR S.P.R. de R.L. de C.V. in Yucatán, Mexico (20°54′31′′ N, 88°11′20′′ W). Following the maturity standards established by Santamaría-Basulto et al. [9], we selected fruits from the middle section of bunches that showed uniform morphology and weighed between 1.0 and 1.2 kg.

The manually harvested fruits were carefully transported to the Scientific Research Center of Yucatán (CICY, A.C.) in Mérida for processing. During transportation, each fruit was protected with double layers of newspaper padding and placed in cardboard boxes lined with cotton cloth to prevent mechanical damage or other mechanical damage. Only specimens free of visible defects or disease symptoms were selected for subsequent experiments. Before experimental use, the fruits underwent a thorough disinfection protocol. After gentle cleaning with a soft sponge, they were immersed in 0.18 mM sodium hypochlorite solution for one minute, then allowed to air-dry under a laminar flow hood to maintain sterile conditions. Experimental design and replicates were conducted as follows: (a) For ripping progression analysis, a total of 24 fruits were evaluated with six treatments, from day 0 (D0) to day 12 (D12), with four fruits for each treatment. And for (b) mechanical damage, the experiment was conducted with treatments D1 and D12 with each group of 24 fruits.

### 2.2. Papaya Ripeness Measurement

Epicarp color (EC) was quantified to assess papaya ripeness following the protocol of Barragán-Iglesias et al. [31]. Ripeness measurements were conducted over several days, with day 0 (D0) marking the start of the experiment and day 12 (D12) as the final treatment day, applying to both the untreated control group and mechanically damaged fruits. Four fruits were used per measurement. EC was measured using a CR-200 chroma meter (Minolta©, Tokyo, Japan) with a 0° viewing angle and 8 mm circumferential illumination, calibrated against a white standard plate (L* = 97.92; a* = −0.45; b* = 2.12). Five readings were taken at three distinct points: the fruit’s center, 5 cm from the stem, and 5 cm from the apex. Color was evaluated using the CIELab* scale, where L* (brightness) ranges from 0 (black) to 100 (white), a* indicates green (−) to red (+), and b* represents blue (−) to yellow (+). Also, the previously defined stages of physiological maturity (RST1) through consumption ripeness (RST5) to over-ripening were used.

### 2.3. Mechanical Damage Treatment

To evaluate the effects of mechanical damage on papaya fruits, the fruits were divided into two groups. The first group included mechanically damaged fruits, while the second was the control group with no mechanical damage. Both groups consisted of fruits at specific ripening stages D1 or D12, with 24 fruits in each group. Mechanical damage was induced using a 10 cm diameter steel plate attached to a Shimadzu AGS-X10kN universal testing machine (Shimadzu Corp.©, Kyoto, Japan) equipped with a calibrated 1 kN load cell. A 50 N compressive force was applied to the equatorial region at a rate of 0.33 mm s^−1^. The force was maintained for 10 s after the plate contacted the fruit. After treatment, the fruits were randomly divided into six batches, with eight fruits in each batch. They were then stored at 25 ± 1 °C under a 12/12 h light/dark cycle and 60–70% relative humidity.

### 2.4. Optical Microscopy Analysis

Histological processing was performed following the protocol outlined by Piven et al. [32]. Briefly, exocarp samples (~1 cm^2^) were collected from the equatorial region of both control and treated fruits at a single ripening stage (either D2 or D12). These samples were fixed in FAA solution (40% formaldehyde, glacial acetic acid, 95% ethanol) for 48 h, followed by dehydration in an ethanol gradient series (50%, 70%, and 100%) and subsequent embedding in JB4 resin. Thin sections (7–10 μm thick) were prepared using a rotary microtome (Thermo Scientific HM 325, Waltham, MA, USA), stained with 0.05% toluidine blue (in 0.2 M buffer), and permanently mounted with Poly/Mount mounting medium (Polysciences Inc., Warrington, PA, USA). Finally, images were captured using a Zeiss Axio Scope A1 microscope (Carl Zeiss, Oberkochen, Germany) equipped with an Axiocam camera.

### 2.5. Protein Extraction Protocol

Proteins were extracted from fruit exocarps using a modified Faurobert et al. method [32]. Three biological replicates (each from four pooled fruits) were processed. Exocarp strips (~0.3 cm thick) were excised from the equatorial region [33], flash-frozen in liquid nitrogen, and homogenized. For extraction, 0.2 g of powder was mixed with 1 mL of ice-cold buffer (500 mM Tris-base pH 8, 50 mM EDTA, 700 mM sucrose, 100 mM KCl, 2% β-mercaptoethanol, 1 mM PMSF, 1% protease inhibitor cocktail). After vortexing and 5 min incubation at 4 °C, 1 mL of Tris-buffered phenol was added, followed by 20 min shaking on ice. The mixture was centrifuged (15,000× *g*, 15 min, 4 °C), and the phenolic phase was collected. Proteins were precipitated overnight at −20 °C with 5 volumes of cold acetone (0.07% β-mercaptoethanol), pelleted (10,000× *g*, 15 min, 4 °C), and washed twice (80% acetone with/without β-mercaptoethanol). Pellets were vacuum-dried, resuspended in PBS/1% SDS, sonicated (20 min), and centrifuged (15,000× *g*, 10 min, 24 °C). Supernatants were stored at −80 °C, and protein concentrations were measured using the Pierce BCA Protein Assay Kit (Thermo Scientific cat 23225). The protocol reported by [34] was followed, and critical steps regarding protein degradation inhibition were considered during the execution of all the steps. For example, first extraction with the Tris buffer contains a protease inhibitor cocktail and the PMSF; also, β-mercaptoethanol at the following steps was added to prevent protein oxidation. Moreover, all the steps during the protocol executions were under ice and controlled temperature to prevent protein degradation.

### 2.6. Digestion and Tandem Mass Tag (TMT) Labeling

For protein digestion, 100 µg of protein aliquots were first reduced with tris-(2-carboxyethyl) phosphine at 60 °C for 45 min, followed by alkylation with 30 mM iodoacetamide (IAA) for 1 h at room temperature in the dark. The reaction was quenched with 30 mM DTT for 10 min before precipitating proteins overnight in cold acetone at −20 °C. After centrifugation (10,000× *g*, 15 min, 4 °C), the pellets were vacuum-dried for 5 min and resuspended in 100 µL digestion buffer (50 mM triethylammonium bicarbonate, 0.1% SDS). Protein digestion was performed using trypsin at a 1:30 (*w*/*w*) ratio overnight at 37 °C, followed by a second digestion with fresh trypsin (1:60 ratio) for 4 h at 37 °C. The reaction was terminated by flash-freezing in liquid nitrogen. For quantitative proteomics, we employed TMT 6-plex isobaric labeling (Thermo Scientific, cat. 90061). Six multiplex strategies currently improve the efficiency and robustness by analyzing up to six samples simultaneously with TMT reagents and with unique reporter ion masses. During the fragmentation step in the apparatus, the previous tagging step improves the protein quantification across samples. Then, comparing the signal and intensities of the ion mass reporters for each sample, a relative quantification can be executed between samples and treatments. Control samples (D2 and D12) were labeled with TMT-126, -127, and -128 tags, while treated samples received TMT-129, -130, and -131 tags. After labeling, all samples were pooled and dried for subsequent analysis.

### 2.7. Protein Fractionation

The labeled peptides underwent two-dimensional fractionation to enhance resolution. First, strong cation exchange (SCX) chromatography was performed using HyperSep SCX cartridges (Thermo Scientific, cat. 60108-421) equilibrated with 5 mM KH_2_PO_4_ in 25% acetonitrile (pH 3.0). Peptides were eluted using a step gradient of increasing KCl concentrations (75 mM, 250 mM, and 500 mM) in the same buffer system. Following SCX separation, all fractions were desalted using HyperSep C18 cartridges (Thermo Scientific, cat. 60108-302). For further resolution, the SCX fractions were subjected to high-pH reversed-phase fractionation using a Pierce kit (cat. 84868) with an eight-step acetonitrile gradient (10–50% in 2.5–25% increments). Finally, all fractions were concentrated by vacuum drying prior to mass spectrometry analysis.

### 2.8. Nano-LC-MS/MS and Synchronous Precursor Selection (SPS)-MS3

The samples were analyzed using an Orbitrap Fusion™ Tribrid™ mass spectrometer (Thermo Fisher Scientific) equipped with an EASY-Spray™ nano-ion source and coupled to an UltiMate 3000 RSLC system (Dionex, Sunnyvale, CA, USA). Each reconstituted sample (50 µL in 0.1% formic acid) was loaded at 0.05 µL·s^−1^ onto a nanoViper C18 trap column (3 µm, 75 µm × 2 cm, Dionex) and separated on an EASY-Spray C18 RSLC analytical column (2 µm, 75 µm × 25 cm) using a 100 min gradient at 5 nL·s^−1^. The mobile phases consisted of 0.1% formic acid in water (solvent A) and 0.1% formic acid in 90% acetonitrile (solvent B). The gradient profile began with 10 min of solvent A, followed by a linear increase to 20% B over 25 min, a 15 min hold at 20% B, a ramp to 25% B over 15 min, a sharp increase to 95% B over 20 min, and a final 8 min re-equilibration with solvent A.

The mass spectrometer operated in positive ion mode with a nanospray voltage of 3.5 kV and a source temperature of 280 °C, using caffeine, MRFA peptide, and Ultramark 1621 (Thermo Fisher Scientific) for external calibration. Full MS scans (380–1500 *m*/*z*) were acquired in the Orbitrap with an AGC target of 4.0 × 10^5^, a maximum injection time of 50 ms, an intensity threshold of 5.0 × 10^3^, a 90 s dynamic exclusion, and a 10 ppm mass tolerance. For MS2 analysis, the top 20 precursors (charge states 2–7) were isolated and fragmented via CID (35% collision energy, activation Q = 0.25) in the ion trap, with an AGC target of 2.0 × 10^4^ and a 50 ms maximum injection time. The precursor range was set to 400–1200 *m*/*z*, with exclusion widths of 18 *m*/*z* (low) and 5 *m*/*z* (high). MS3 spectra were acquired using synchronous precursor selection (SPS) of 10 notches, with HCD fragmentation (65% collision energy) and Orbitrap detection (60,000 resolution, 120–500 *m*/*z* range, AGC 1.0 × 10^5^, 120 ms max injection time, 1 microscan) [35,36]. The raw data were deposited in the iProX repository under accession PXD028076.

### 2.9. Protein Identification and Quantification

The raw mass spectrometry data were processed using Proteome Discoverer 2.1 (PD, Thermo Fisher Scientific Inc.), which incorporated three distinct search algorithms: Mascot (v.2.4.1, Matrix Science, Boston, MA, USA), SEQUEST HT 2.4 [35], and MS AMANDA 2.0 [37]. Database searches were performed against the *Carica papaya* (NCBI: txid3649) proteome, obtained from the non-redundant NCBI database. The search parameters specified full-tryptic digestion with a maximum of two missed cleavages permitted.

Static modifications were defined as TMT 6-plex labeling (+229.163 Da) on N-terminal and lysine residues and carbamidomethylation of cysteine (+57.021 Da). Dynamic modifications included methionine oxidation (+15.995 Da) and deamidation of asparagine and glutamine residues (+0.984 Da).

Peptide identification was conducted in the linear ion trap at low resolution, employing mass tolerances of ±10 ppm for precursor ions and ±0.6 Da for fragment ions. To ensure high-confidence matches, peptide spectral matches (PSMs) were filtered at a 1% false discovery rate (FDR) using the Percolator algorithm [38].

For quantitative analysis, the TMT 6-plex reporter ion method was applied, utilizing the most accurate centroid peaks with a ±10 ppm mass tolerance and a 45% precursor co-isolation threshold. When employing the SPS-MS3 acquisition strategy, quantification was performed at the MS3 level, and only spectra containing all six reporter ions were classified as quantifiable.

At the protein level, protein grouping was enabled, and abundances were determined based on the median of all quantifiable PSMs corresponding to each protein group (master proteins). To proceed with the formal and statistical analysis, a normalization of median approach was conducted, and the TMT ratios were transformed to log2 before statistical analysis and plotting. Finally, normalized TMT ratios were computed relative to the total signal of all reporter ions within each spectrum.

### 2.10. Bioinformatic Analysis

As a non-model species, the annotation and interaction analysis of papaya proteins rely on the search tools and customized databases employed. To address this, we first annotated all identified proteins using the NetGO 2.0 web service [39], a tool that enhances large-scale automated function prediction (AFP) by integrating extensive protein–protein network data. Here, we observe that secondary analysis by means of BLASP (v 2.16.0) or InterProScan v.5.75-106.0 could improve our functional and homology identity assignation. However, this study presents the annotation data with NetGO v.2.0 as a starting point of functional annotation. Further analysis should be carried out with updated databases.

For Gene Ontology (GO) enrichment analysis, we analyzed four sets of differentially abundant proteins (DAPs)—D2-UP, D2-DOWN, D12-UP, and D12-DOWN—using the topGO package (https://bioconductor.org/packages/release/bioc/html/topGO.html, accessed on 15 May 2024) from Bioconductor. The analysis was performed using Fisher’s exact test with the “classic” algorithm, considering only Biological Process GO terms with a confidence score of ≥0.6.

Protein–protein interaction (PPI) networks were retrieved and analyzed using the STRING database (v11). To ensure valid STRING identifiers, we first matched the identified papaya proteins against the entire predicted papaya proteome (Phytozome v13, Cpapaya_113_ASGPBv0.4). The STRINGdb package (https://www.bioconductor.org/packages/release/bioc/html/STRINGdb.html, accessed on 20 July 2024) in R was used to generate independent PPI networks for D2 and D12, applying a default confidence threshold of 400 (medium confidence). Additionally, STRINGdb facilitated the identification of protein clusters and KEGG pathways. Networks were visualized using Gephi v0.9.2 and Igraph v.2.1.4.

For pathway analysis, we calculated the log2 fold change ratios of differentially abundant proteins for both D2 and D12. These data were then processed using the Plant Metabolic Network (PMN) Omics Viewer (https://pmn.plantcyc.org/organism-summary?object=PAPAYA, accessed on 3 May 2024), leveraging the Carica papaya database (Cpapaya_113_ASGPBv0.4.protein.A, https://plantcyc.org/, accessed on 3 May 2024). A differential pathway perturbation score (DPP) threshold of 50 was applied to assess pathway alterations between control and damaged conditions at D2 and D12. Although the threshold could be arbitrary, our principal reason to use this value was for easily distinguishing between significant and insignificant enriched pathways. Nevertheless, a more commonly used threshold is above 70%. The results were used to generate pathway diagrams and protein accumulation heatmaps.

### 2.11. Statistical Analysis

Statistical analysis was performed using the DEP Bioconductor package [40], with non-normalized protein abundances as input. Differentially abundant proteins (DAPs) were defined as those with adjusted *p*-values (*p*-adj.) <0.05 and a fold change (FC) threshold of ≥1.4. For EC, multiple pairwise comparisons were conducted using Tukey’s test (α = 0.05) in R to evaluate differences between groups for each parameter. Here, we state that the biological and technical replicates were used as follows: (a) For ripping progression analysis, a total of 24 fruits were evaluated with six treatments, from day 0 (D0) to day 12 (D12), with four fruits for each treatment. And (b), for mechanical damage, the experiment was conducted with treatments D1 and D12 with each group of 24 fruits.

## 3. Results

### 3.1. Epicarp Color Measurement

We examined the damage caused by applied pressure to papaya fruit by first recording color changes in damaged papaya fruit compared to control samples. The CIE parameters (L*, a*, and b*) increased over time (Figure 1A). In non-damaged papaya fruit, brightness (L*) ranged from 44.9 ± 1.1 (D0) to 55.5 ± 4.45 (D12), redness (a*) from −14.7 ± 1.5 to 6.96 ± 3.71, and yellowness (b*) from 22.9 ± 1.2 to 42.2 ± 5.61. For damaged fruit, minimum values were identical to controls, while maximum values were 54.1 ± 5.63, 8.42 ± 3.59, and 31.3 ± 2.97 for L*, a*, and b*, respectively. The b* values (yellowness) were higher in non-damaged fruit (Figure 1B).

### 3.2. Assessment of Physical Damage Using Optical Microscopy

Structural alterations induced by mechanical damage were evaluated through detailed histological analysis. At D2, non-damaged control fruit exhibited characteristic isodiametric epidermal cells and turgid, often bulliform, hypodermal cells, with cell size progressively increasing in deeper cortical layers. In stark contrast, mechanically damaged fruit at D2 displayed epidermal cells that adopted a compressed, rectangular morphology, a characteristic also observed in the initial hypodermal cell layers.

By D12, control papaya fruit presented an epidermis exhibiting a thickened cuticle, and the subsequent 5–8 hypodermal cell layers displayed evidence of suberization and plasmolysis. Parenchyma cells exhibited a loosely rounded morphology with a discernible reduction in cell wall thickness. In contrast, mechanically damaged tissues at D12 revealed severely compacted and suberized epidermal and hypodermal cells, with suberin deposition extending remarkably deep, to approximately the 50th cell layer. Underlying parenchyma cells exhibited highly sinuous and plasmolyzed cell walls, accompanied by the formation of notably enlarged intercellular spaces (Figure 2C).

### 3.3. Protein Extraction Protocol

To confirm the integrity and successful extraction of total proteins from papaya exocarp, samples from control and mechanically damaged fruits at D2 and D12 were analyzed by one-dimensional sodium dodecyl sulfate-polyacrylamide gel electrophoresis (1D SDS-PAGE). The electrophoretic profiles, visualized by SYPRO^®^ Ruby staining, revealed distinct protein bands across all tested samples, indicating a robust and reproducible protein extraction protocol. The presence of a wide range of protein molecular weights, without significant degradation or smearing, confirmed the high quality of the extracted proteins suitable for downstream proteomic analysis (Appendix A).

### 3.4. Bioinformatics Analysis

Principal component analysis (PCA) was performed on the proteomic results to elucidate major sources of variation. The data exhibited clear clustering based on treatment, with the first two principal components (PC1 and PC2) collectively explaining 98.8% of the total variance (PC1: 96.5%; PC2: 2.3%) (Figure 3A), indicating distinct proteomic profiles induced by mechanical damage. High repeatability was observed among biological replicates within each treatment group, evidenced by strong correlation coefficients (Figure 3B), and the consistency of differentially abundant proteins (DAPs) across replicates was robust (Figure 3C).

At day 2 (D2) post-treatment, 2465 proteins were identified, of which 165 were classified as DAPs, including 104 up-regulated and 61 down-regulated proteins. By day 12 (D12), the total number of identified proteins increased to 2857, with 536 DAPs comprising 342 up-regulated and 194 down-regulated proteins (Figure 3D). Across both time points, 3230 unique proteins were identified. Comparative analysis revealed that 23 up-regulated proteins were common to both D2 and D12, while 10 down-regulated proteins were shared between time points. Notably, eight up-regulated proteins at D2 exhibited down-regulation at D12, and only one down-regulated protein at D2 showed up-regulation at D12.

Gene Ontology (GO) enrichment analysis revealed that mechanical damage generally decreased proteins associated with photosynthesis, light and radiation responses, and abiotic stimuli (up to *p* ≥ 7 × 10^−7^). Conversely, proteins involved in biotic response and defense mechanisms (*p* ≥ 0.0005) were consistently induced at both D2 and D12 (Figure 4A,B).

Protein–protein interaction (PPI) analysis provided further insights into the functional networks affected by mechanical damage. At D2, up-regulated DAPs formed a network comprising 102 nodes and 7 edges, though these interactions were not statistically significant (*p* > 0.05). In contrast, down-regulated DAPs at D2 clustered into four distinct groups with 60 nodes and 141 edges, showing significant enrichment (*p* < 0.05). At D12, up-regulated DAPs formed eight clusters with significant PPI enrichment (336 nodes and 77 edges), while down-regulated DAPs comprised 188 nodes and 264 edges (Figure 5A).

Kyoto Encyclopedia of Genes and Genomes (KEGG) enrichment analysis of protein networks corroborated these findings. In the D2 network, cluster c1 showed significant enrichment of proteins associated with primary energy metabolism. Clusters c2, c3, and c4 were enriched with proteins involved in translation, plant hormone and plant–pathogen interaction, photosynthesis, and carbon fixation metabolism. For the D12 network, cluster c2 was enriched in pathways related to carbon metabolism, photosynthesis, energy metabolism, and the biosynthesis of secondary metabolites. Cluster c1 was predominantly enriched in translation and genetic information processing proteins. Cluster c3 contained proteins associated with amino acid and secondary metabolism, while cluster c4 was enriched in vesicular trafficking and plant–pathogen interaction proteins (Figure 5B,C). These KEGG findings aligned well with GO enrichment analysis observations.

Mechanical damage profoundly affected specific metabolic pathways. Photosynthesis was severely impacted, evidenced by down-accumulation of four photosystem-related proteins at D2 (photosystem II 22 kDa protein, photosystem I subunit III, photosystem I subunit V, and photosystem I subunit X) and eight at D12 (photosystem II 22 kDa protein, photosystem II 13 kDa protein, photosystem I subunit V, plastocyanin, cytochrome b6-f complex subunit 4, ferredoxin, F-type H+/Na+-transporting ATPase subunit alpha [EC:7.1.2.2 7.2.2.1], and F-type H+-transporting ATPase subunit gamma) (Figure 5B). At D2, four antenna proteins (LHCA1, LHCA2, LHCA3, and LHCA4) of the light-harvesting chlorophyll protein complex declined, though these proteins remained unaffected at D12 (Figure 5B,C). Conversely, chlorophyllase-2, Calvin cycle protein CP12-3, and ribulose bisphosphate carboxylase small chain 1 proteins were induced, while ribulose bisphosphate carboxylase/oxygenase activase content was reduced (Figure 5B).

In glycolysis/gluconeogenesis, down-accumulation of fructose-1,6-bisphosphatase I [EC:3.1.3.11] and NADP-dependent glyceraldehyde-3-phosphate dehydrogenase isoform 1 [EC:1.2.1.9] occurred at D2. At D12, a more extensive down-accumulation affected eight proteins: fructose-1,6-bisphosphatase I [EC:3.1.3.11], fructose-bisphosphate aldolase class I [EC:4.1.2.13], glyceraldehyde-3-phosphate dehydrogenase (NADP+) [EC:1.2.1.9], dihydrolipoamide dehydrogenase [EC:1.8.1.4], alcohol dehydrogenase 1/7 [EC:1.1.1.1], alcohol dehydrogenase (NADP+) [EC:1.1.1.2], glucose-6-phosphate 1-epimerase [EC:5.1.3.15], and aldose 1-epimerase [EC:5.1.3.3] (Figure 5B,C). Furthermore, a reduction in the levels of several ribosomal proteins was detected at D12, including S5, S8, S10, S27, and L32, while at D2, proteins S5, L18, L32, S1, and S28 were down-accumulated (Figure 5B,C), suggesting an impact on protein synthesis machinery.

Regarding secondary metabolism, key enzymes in phenylpropanoid metabolism were reduced at D12, specifically phenylalanine ammonia-lyase [EC:4.3.1.24], ferulate-5-hydroxylase, and shikimate O-hydroxycinnamoyl transferase (Figure 5C). Other enzymes showing altered accumulation included 15-cis-zeta-carotene isomerase and xanthoxin dehydrogenase, both involved in carotenoid breakdown. Consistent down-accumulation of a peroxidase was noted at both D2 and D12 (Figure 5B,C).

Proteins associated with cell wall regulation, callose metabolism, and plasmodesmata regulation were also significantly affected (Figure 5B,C). At both D2 and D12, plasmodesmata callose-binding protein 3 was up-accumulated. At D12, additional induction included plasmodesmata callose-binding protein 5, callose synthase 10, and callose synthase 2. At D2, glucan endo-1,3-beta-glucosidase and xyloglucan endo-transglycosylase accumulated, with glucan endo-1,3-beta-glucosidase-like isoform 1 also accumulating at D12. Additionally, at D2, proteins involved in cell wall biosynthesis and regulation, including secoisolariciresinol dehydrogenase, cinnamoyl-CoA reductase, leucine-rich repeat extensin-like protein 3, and xyloglucan endo-transglycosylase, showed accumulation. Titin-like protein accumulation was consistent at both D2 and D12, as were salicylate carboxy methyltransferase, PLAT domain-containing protein 3, endochitinase, and dnaJ protein homolog ANJ1.

The study also revealed accumulation of pathogenesis-related (PR) proteins and those involved in biotic and abiotic stress responses, including RPM1-interacting protein 4, pathogenesis-related protein 1, SERF-like protein, BS domain-containing protein, respiratory burst oxidase homolog protein D, and glucan endo-1,3-beta-glucosidase A6. Furthermore, proteins involved in calcium transport and regulation, such as sodium/calcium exchanger NCL2 and calmodulin-like protein 8, showed altered accumulation (Figure 5B,C). At D12, significant induction of various proteases (e.g., papaya proteinase omega, papain, and chymopapain or chymopapain isoform III) and protein inhibitors (e.g., cysteine proteinase inhibitor, subtilisin-like serine protease inhibitor, cysteine proteinase inhibitor A, late serine proteinase inhibitor, and cystatin) was observed. Up-regulation of numerous calmodulins and protein kinases, as well as calreticulin and calcium-transporting ATPase 4 endoplasmic reticulum, indicated active signaling and calcium regulation at D12. Similar accumulation patterns were observed for other proteins involved in biotic and abiotic stress responses, including Chain A Papaya Barwin-like protein, universal stress protein PHOS32, hevamine-A, elicitor-responsive protein 1-like, and thaumatin-like protein.

Plant hormone metabolism was also notably affected by mechanical damage (Figure 6). Cytokinin O-glucoside biosynthesis was activated at D12, while biosynthetic pathways for abscisic acid, cytokinin 7-N-glucosides, and flavonol glucosylation were repressed in both treatments. Conversely, ethylene and gibberellin biosynthesis pathways showed activation in both treatments. At D2, S-adenosyl methionine synthetase (SAM synthetase) and 1-aminocyclopropane-1-carboxylate oxidase were over-accumulated, while at D12, both S-adenosyl homocysteine hydrolase and 1-aminocyclopropane-1-carboxylate oxidase were down-accumulated.

A complete list of all identified proteins and their respective abundances at D2 and D12 is provided in Appendix A.

## 4. Discussion

### 4.1. Exocarp Color as Papaya Ripeness Indicator

Accurate and non-destructive assessment of physiological maturity is paramount for effective postharvest management and quality control of perishable fruits. In this study, we employed a comprehensive non-destructive methodology for evaluating papaya physiological maturity, proving particularly valuable for the subsequent assessment of mechanical damage. This approach builds upon the established work of Barragán-Iglesias et al. [31], who demonstrated that physicochemical changes during ripening define the developmental stages of the “Maradol” papaya, from physiological maturity (RST1) through consumption ripeness (RST5) to over-ripening (ORF). Their research notably established strong correlations between destructive parameters (e.g., mesocarp firmness, total soluble solids [TSS]) and non-destructive indicators, such as epicarp color measured by the CIELab scale, enabling the development of predictive models for on-tree maturity estimation. Our findings not only corroborate these established relationships but also crucially refine them by establishing stage-specific thresholds under natural growing conditions. This advancement carries significant industrial implications, including improved fruit sorting for diverse markets (fresh, export, or processing), optimized harvest timing, enhanced sensory quality, and extended postharvest shelf life. These factors are critical for the effective management of mechanically stressed papayas.

### 4.2. Proteomic Insights into Papaya Exocarp Response to Mechanical Damage

Proteomic studies have greatly enhanced our understanding of fruit ripening by uncovering key molecular mechanisms [18,41,42,43,44,45,46,47,48,49]. However, proteomic changes induced by mechanical damage, particularly in papayas, remain poorly understood. This study provides the first detailed proteomic profile of *Carica papaya* exocarp in response to mechanical damage, a tissue crucial for fruit quality and postharvest shelf life [12,50,51]. Our comparative analysis of two postharvest time points reveals a ripening-stage-dependent proteomic response to mechanical stress. Figure 7 summarizes the key findings. In the following sections, a more detailed and extended discussion about topics like the mechanical changes at the proteomic level and their association with secondary metabolism will be founded. However, it is an intriguing observation that other works and studies had previously shed light on and discussed the changes at metabolite and protein levels with a combination of metabolomics and proteomic approach. Some works summarizing this relationship are described in [52,53]. A recent study by [54] investigated the metabolomic and protein level changes in leaf tissues of spring wheat subjected to drought stress. The authors showed a variant susceptible to drought and another drought-resistant variant. Drought stress caused a severe decrease in protein content in the drought-susceptible variant, with increasing levels of aromatic and branched chain amino acids, particularly tryptophan accumulation, related to auxin production. On the other hand, the drought-resistant variant showed minor changes in protein levels and metabolomic approach levels, with only purine metabolism having significant changes observed. The authors concluded that these insights can be used to study important molecules and can be used to improve crop management.

### 4.3. Mechanical Damage Triggers Ripening-Stage-Specific Proteomic Remodeling in Papaya Exocarp

Mechanical injury is a major stressor for climacteric fruits like papaya during postharvest handling. Despite its economic impact, the molecular basis of papaya’s response to mechanical damage remains unclear. Using tandem mass tag quantitative proteomics, we profiled the exocarp proteome at two ripening stages—day 2 (D2) and day 12 (D12) postharvest—following mechanical damage. Contrary to being a passive barrier, the exocarp emerges as a dynamic tissue orchestrating complex wound responses. Our data show that papaya modulates distinct sets of proteins depending on the ripening stage, shifting from early signaling and metabolic adaptation to late-stage structural reinforcement and defense deployment.

### 4.4. Calcium and ROS: Key Messengers in Damage Perception and Signal Amplification

Tissue disruption rapidly activates exocarp perception mechanisms, where calcium ions (Ca^2+^) and reactive oxygen species (ROS) play central roles. At both D2 and D12, we observed up-regulation of proteins associated with calcium signaling, including calmodulins, Ca^2+^-ATPases, and calreticulins—key molecules that decode Ca^2+^ transients into biochemical responses [55,56,57,58]. Calmodulins, in particular, act as versatile sensors, interacting with proteins involved in transcriptional regulation, hormone signaling, and cytoskeletal remodeling [59]. The presence of Ca^2+^-ATPases, which sequester cytosolic Ca^2+^ into organelles like the endoplasmic reticulum, indicates tight Ca^2+^ homeostasis regulation during wound signaling [60].

Notably, at D2, we detected the accumulation of respiratory burst oxidase homolog D (RbohD), a membrane-bound NADPH oxidase that generates ROS in response to mechanical and biotic stress [61]. These ROS reinforce cell walls, act as antimicrobials, and serve as secondary messengers, activating downstream pathways like MAPK cascades and hormone signaling [62]. Together, these findings highlight calcium and ROS as an evolutionarily conserved early-response module in fruit wound signaling.

### 4.5. Ethylene Biosynthesis Modulation: A Ripening-Dependent Response

Ethylene, a key regulator of climacteric fruit ripening and senescence, is also integral to stress responses [63,64]. Our data revealed nuanced regulation of 1-aminocyclopropane-1-carboxylic acid oxidase (ACO), the terminal enzyme in ethylene biosynthesis [65]. ACO abundance increased at D2 post-wounding, suggesting a transient ethylene surge to coordinate defense and repair [66]. Conversely, ACO declined at D12, aligning with reduced yellowness in bruised fruits, possibly delaying ripening. This differential regulation mirrors findings in tomato and papaya, where stress modulates ripening via hormonal crosstalk [26,67]. The D12 decline may prevent excessive softening, prioritizing tissue integrity over maturation. Plant hormone metabolism and signaling pathways, like the ethylene route for example, can explain the observed phenotype of ripened fruits, as climacteric ethylene production modulates the activity of several cell-wall-modifying enzymes through the activity of master regulators [68].

### 4.6. ABA and Carotenoid Biosynthesis Are Suppressed at Late Ripening Stages

In addition to ethylene, abscisic acid (ABA) plays a critical role in fruit ripening, promoting sugar accumulation and pigment biosynthesis. At D12, we observed reduced abundance of xanthoxin dehydrogenase (XanDH), an enzyme involved in ABA biosynthesis [69,70]. Similarly, 15-cis-ζ-carotene isomerase (Z-ISO), essential for carotenoid precursor synthesis, was down accumulated [71,72]. This suppression suggests that mechanical damage at late stages diverts metabolic flux away from pigment and hormone production, impairing color development and sweetness [61]. More intriguingly, the crosstalk between ABA and ethylene signaling pathways has been described. Although ABA signaling regulating ripening is frequently associated with non-climatiric fruits, and ethylene with climatiric fruits such as papaya, there is evidence supporting the connection between both pathways regarding the fruit’s nature. This crosstalk has been evidenced in tomato, where SINAC1 regulates the expression of downstream factors, therefore integrating both signaling pathways to balance fruit ripening [73]. Reduced ABA may further limit ethylene production, compounding ripening delays [74].

### 4.7. Photosynthesis and Primary Metabolism Are Down-Regulated During Late Defense Activation

Although papaya fruit retains active chloroplasts in early ripening [75], damage at D12 broadly repressed photosynthesis-related proteins, resembling herbivore/pathogen responses in leaves, where photosynthesis is suppressed in favor of activating defense pathways [76,77]. Glycolysis/gluconeogenesis enzymes also declined, indicating a metabolic slowdown. This likely reflects resource reallocation from energy production to structural defense—a common stress response [26]. Although the critical roles of the primary metabolism can be affected through the ripening process, there are still areas that need to be explored in order to clarify how other aspects of primary metabolism are affected. For example, our data suggest the regulation of AtRPS6A, an important component of the 40S small ribosomal subunit, which in turn controls the cell growth and more intriguingly the selective translation of specific mRNA through several physiological conditions, including physical damage [78]. The regulation of this key player could improve responses to mechanical damage.

### 4.8. Plasmodesmata Regulation via Callose Metabolism: Balancing Connectivity and Defense

Plasmodesmata are microscopic channels that traverse plant cell walls, allowing direct cytoplasmic continuity between adjacent cells for the transport of nutrients, signaling molecules, and RNAs. Their permeability is tightly regulated under stress [79]. A major regulator of plasmodesmal conductance is callose, a β-1,3-glucan polymer deposited at the neck region of the channel [80]. At D12, callose synthase accumulation suggested increased callose deposition and plasmodesmata closure, limiting damage spread [81,82]. In contrast, β-1,3-glucanases (which degrade callose) were up-regulated at D2, enhancing symplastic transport. This dynamic regulation reflects a stage-specific balance between metabolic connectivity and physical barrier reinforcement.

### 4.9. Suberization: A Lipid-Based Defense Against Water Loss and Pathogens

At D12, proteins linked to suberization, including cytochrome P450s, ABC transporters, and lipid transfer proteins were up-accumulated. Suberin is a complex polyester of fatty acids and phenolics deposited in cell walls, known for creating hydrophobic barriers in response to wounding [82,83]. Its deposition is a hallmark of secondary defense, physically sealing off damaged areas to prevent desiccation and pathogen entry [83,84]. In papaya exocarp, suberization may complement cuticle repair (Figure 2).

### 4.10. Induction of Pathogenesis-Related Proteins and Vesicular Transport

At D12, pathogenesis-related (PR) proteins, including chitinases and β-glucanases, accumulated significantly. These proteins are central to defense responses, likely induced by hormones such as salicylic acid (SA), jasmonic acid (JA), and ethylene, and perform antimicrobial, cell wall remodeling, and signaling functions [75,76]. Concurrent up-regulation of vesicle trafficking proteins (e.g., SNAREs) suggests active PR protein secretion via the endomembrane system [77,79], highlighting the exocarp’s role as a responsive defense front.

### 4.11. Challenges and Future Directions

Plants confront mounting challenges in adapting to climate change, requiring sophisticated mechanisms to respond to increasingly hostile environmental conditions. Although current proteomic studies address the fundamental questions about the changes and the nature of the proteins and their abundance due to different physiological processes, other aspects, such as the influence of post-translational modifications (PTM) and how these can regulate enzyme or protein activities, remain elusive [85,86]. Current research has significantly advanced our understanding of how PTMs regulate stress-responsive gene expression and critical physiological processes. While substantial progress has been made in characterizing PTM functions in mechanical stress responses, critical gaps remain in our understanding of PTM-mediated adaptive mechanisms. The principal obstacle stems from the inherent proteome complexity, where PTMs generate an expansive repertoire of proteoforms, structurally and functionally distinct protein variants originating from identical genetic sequences. These proteoforms emerge through precise combinatorial patterns and spatial arrangements of PTMs, creating a functional diversity that currently defies complete characterization and obscures their precise contributions to plant stress adaptation [29]. Therefore, the study of how and what type of proteoforms are enriched in plant development could shed light on the molecular basis of processes with limited access. Other areas and subjects to study include phytohormone biosynthesis, photosynthetic efficiency, carbon assimilation, primary metabolic pathways, and stress signal transduction—all essential for rapid environmental adaptation with which to reconcile proteomic information rising on several plant developmental stages and processes such as ripening.

## 5. Conclusions

This study provides the first in-depth characterization of the proteomic remodeling triggered by mechanical damage in postharvest papaya (*Carica papaya*) fruit, revealing a complex and ripening-stage-dependent response in the exocarp. Early in ripening, the fruit prioritizes rapid signaling, hormonal modulation, and metabolic adjustments to perceive and adapt to injury. In contrast, advanced ripening stages activate robust structural defenses, including suberization, callose deposition, and the accumulation of pathogenesis-related proteins. These findings demonstrate that the papaya peel is not a passive protective layer but a highly responsive and metabolically active tissue, fine-tuned to maintain fruit integrity under stress.

By uncovering the molecular architecture underlying the papaya’s response to mechanical damage, this work lays a foundational framework for understanding how mechanical stress impacts climacteric fruit physiology. These insights hold the potential for guiding postharvest technologies and breeding strategies aimed at improving mechanical resilience, extending shelf life, and reducing postharvest losses not only in papaya but potentially in other economically important soft fruits.

## Figures and Tables

**Figure 1 proteomes-13-00044-f001:**
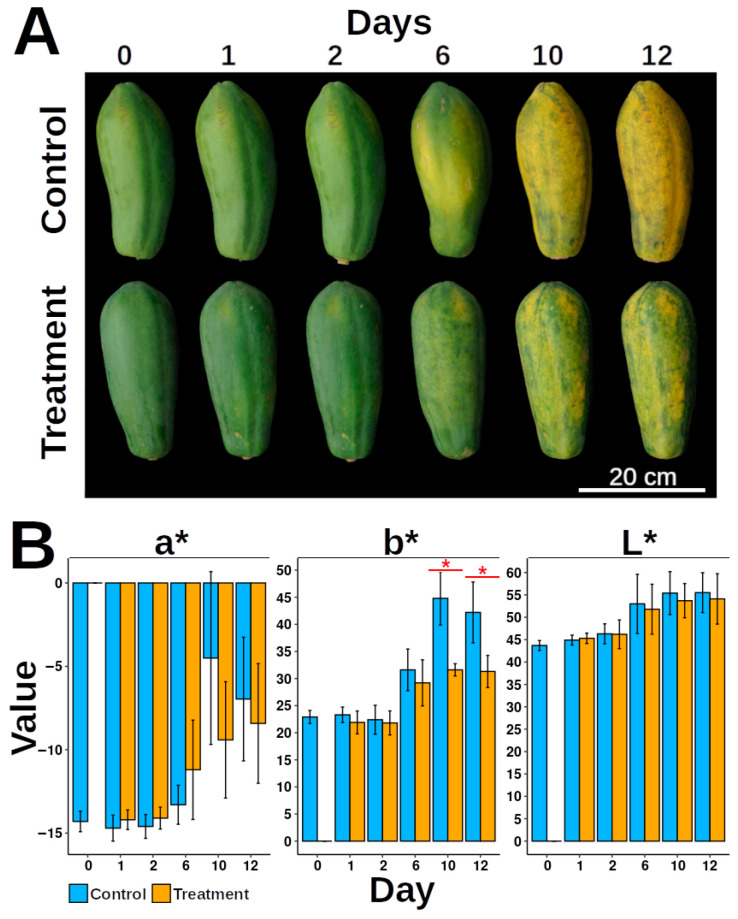
Effect of mechanical damage on papaya fruit ripening. (**A**) Ripening progression of papaya fruits over time. Control: undamaged fruits; Treatment: mechanically damaged fruits. Time points: 0D (day zero), 1D (day one), 2D (day two), 6D (day six), 10D (day ten), and 12D (day twelve). (**B**) Peel color parameters (L*, a*, and b* values) during ripening. Data are presented as means with error bars representing ± standard deviation (*n replicates per group). Red asterisks indicate statistically significant differences (*p* < 0.05, Tukey’s test). The number of experimental replicates is specified in the Materials and Methods Section, Papaya Ripeness Measurement and Mechanical Damage Treatment. Four fruits were used per treatment, totaling *n* = 24 fruits for the experiment.

**Figure 2 proteomes-13-00044-f002:**
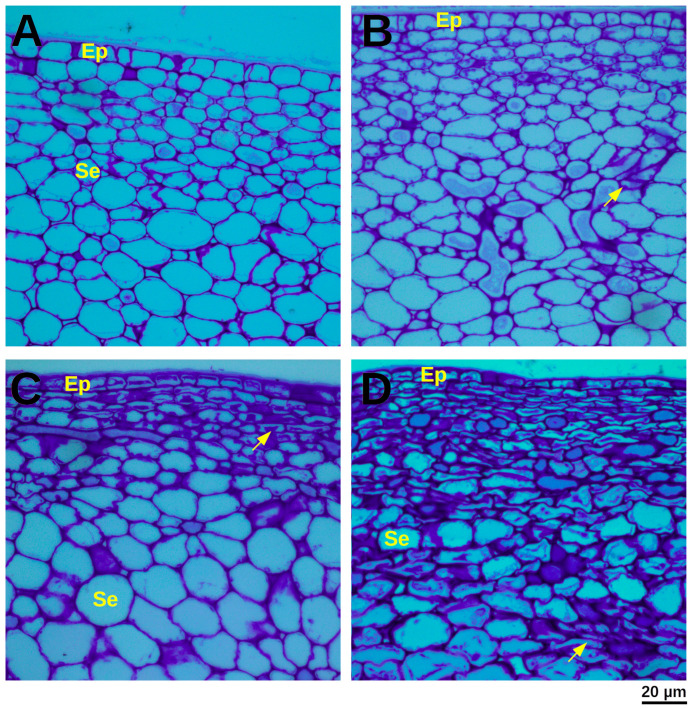
Micrographs of papaya exocarp. (**A**) Undamaged exocarp (D2); (**B**) Undamaged exocarp (D12); (**C**) Mechanically damaged exocarp (D2); (**D**) Mechanically damaged exocarp (D12). EP: epidermis; Se: subepidermis. The solid yellow arrow indicates the cell wall.

**Figure 3 proteomes-13-00044-f003:**
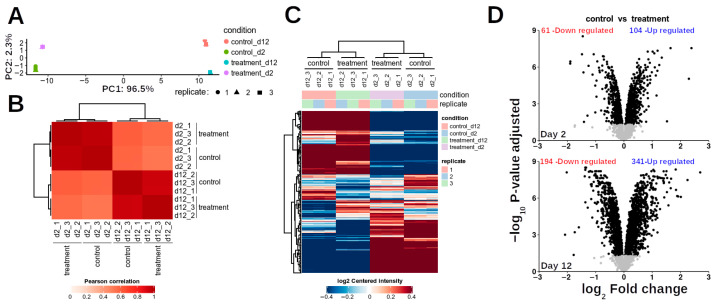
General statistics of identified proteins. (**A**) PCA analysis of protein abundances per condition. (**B**) Pearson’s correlation analysis of protein abundances. (**C**) Heatmap of differentially abundant proteins (DAPs). (**D**) Volcano plot at D2 and D12, showing non-significant proteins (*p* > 0.05) and DAPs (*p* < 0.05, black dots).

**Figure 4 proteomes-13-00044-f004:**
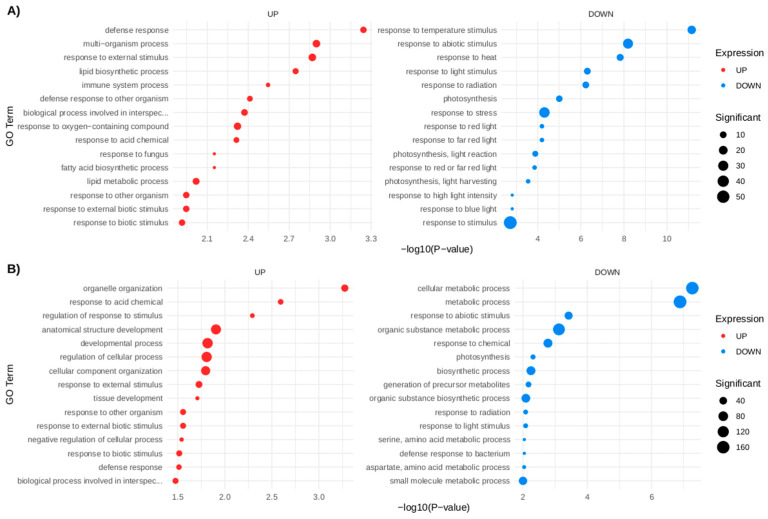
Gene Ontology (GO) enrichment analysis of up- and down-accumulated proteins at (**A**) D2 and (**B**) D12.

**Figure 5 proteomes-13-00044-f005:**
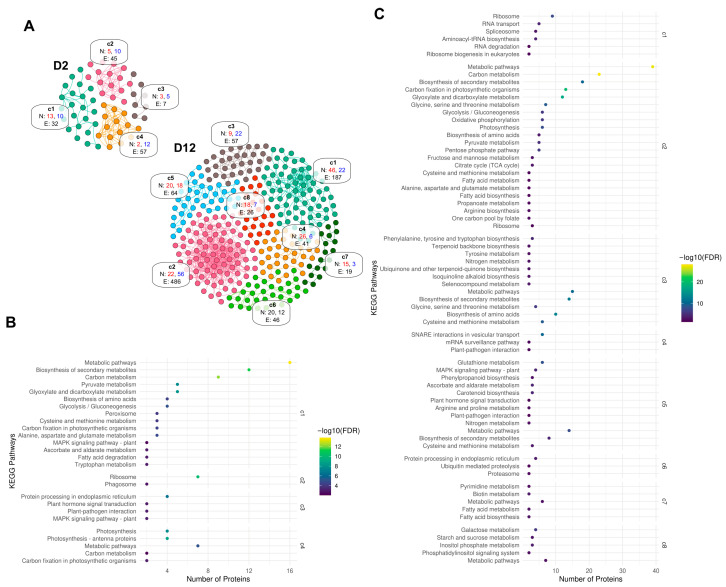
Protein–protein interaction (PPI) networks of DAPs. (**A**) Major interaction clusters at D2 (top left) and D12 (bottom right). Node (N) and edge (E) counts are indicated. Red numbers: up-regulated proteins; blue numbers: down-regulated proteins. Clusters were denoted as c1 to c8, in bold. In cluster c6 the code color was omitted. (**B**) KEGG enrichment of major clusters at D2. (**C**) KEGG enrichment of major clusters at D12.

**Figure 6 proteomes-13-00044-f006:**
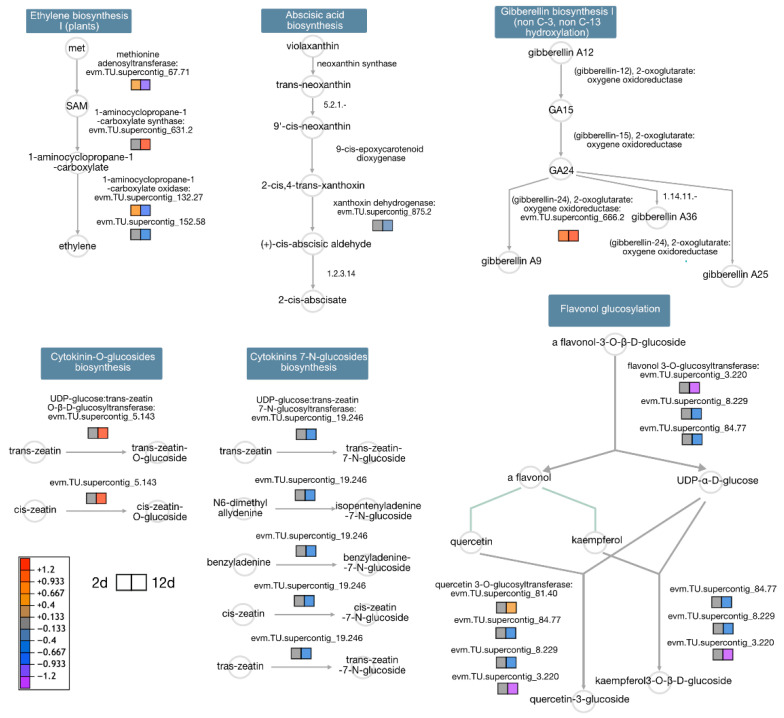
Schematic representation of key plant hormone biosynthetic pathways affected by mechanical damage. Fold change (FC) is expressed as log_2_ (treatment/control) for D2 (left square) and D12 (right square).

**Figure 7 proteomes-13-00044-f007:**
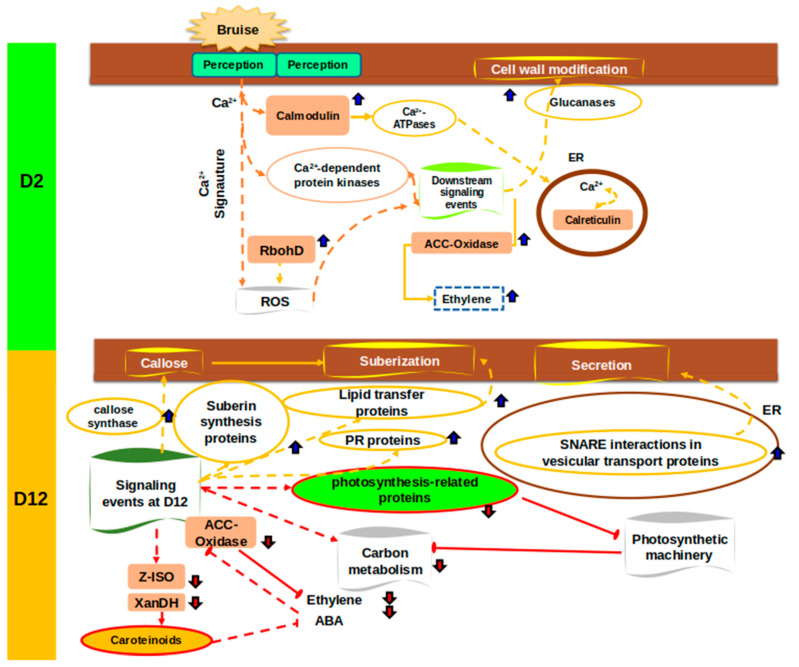
Proposed model of major changes occurring at day 2 and 12 post-treatment. Blue arrows indicate induced processes; red arrows indicate repressed processes.

## Data Availability

Protein identification and quantification data are provided in the accompanying CSV files. Raw data are available via the ProteomeXchange Consortium (iProx repository, ID: PXD028076; https://www.iprox.cn/page/PSV023.html?url=1643232314543wR8I, accessed on 15 May 2024; password: fMyg).

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
