# Peer review of "Proteomic Analysis of Mechanical Injury Effects in Papaya Fruit at Two Maturity Stages"

_proteomes, 2025, doi:10.3390/proteomes13030044_

Round 1
Reviewer 1 Report
Comments and Suggestions for Authors
The manuscript entitled "Proteomic analysis of mechanical injury effects in papaya fruit at two maturity stage" authored by Reyes-Soria et al investigated the proteomics change upon mechanical damage on papaya at two different ripening stages using TMT labeling, and bioinformatics analysis including GO, KEGG pathway, protein-protein interaction were conducted on DAPs identified from the study, providing a proposed change of papaya upon mechanical damage at molecular level.
The methods described and data presented are clear, discussion is in-depth with detailed data interpretation. However, there are a few comments:
- In the abstract, line 21 "the molecular response to fruit bruising remains poorly understood." I think it needs to be rephrased, as there are multiple published/proposed "molecular response" to fruit bruising.
- Line 157, under 2.6, should it IAA?
- What's the proteome coverage and proteome sequence coverage from the current proteomics workflow in this study? Has RTS (Real time search) MS3 been compared with SPS MS3?
- Line 256 and throughout the manuscript, is it differentially abundant proteins instead of differentially accumulated proteins (DAPs)?
- Line 258, either <-1.4?
- Page 8, figure 3A could be bigger in size, it's hard to see all the data points without zooming in significantly.
- Page 8, annotation on figure 3D is up and down "regulated" while in line 317 it's "accumulated".
- Line 319, "342 up-accumulated" while it's 341 up regulated in figure 3D.
- For figure 3, maybe a set of venn diagram to show the overlap and uniqueness between D2/D12 proteins (total, down regulated, up regulated) would be more intuitive.
- Page 10, figure 5A, what does the black numbers in c6 in D12 represent?
- Line 432-434, it seems a redundant copy from guideline?
- Figure 7, I like the idea of proposing a model based on proteomics changes, but is there any metabolomics data to backup the proposed model in similar study as key pathways are heavily metabolites involved?
- Line 551-552, references are needed for the description of how PTMs are involved in stress-responsive gene expression.
- Same paragraph under "Challenges & Future Directions", I think the future directions need to be pointed out in a more clear way.
Reviewer 2 Report
Comments and Suggestions for Authors
Proteomics-3751064: This manuscript presents a comprehensive proteomic investigation into the response of papaya fruit exocarp to mechanical damage at two ripening stages. Using tandem mass tag quantitative proteomics, the authors identify differentially accumulated proteins and analyze associated pathways and defense mechanisms.
L3: Please fix “Pro-teomic” to “Proteomic”. Remove the hyphenation and check throughout the manuscript.
L25–29: The term “up-accumulated” and “down-accumulated” is used repeatedly. Please consider using “upregulated” and “downregulated” for consistency with broader proteomics literature.
L36–39: Please consider citing more recent reviews or global data on papaya postharvest losses to strengthen context.
L90–100: Fruit selection and handling are described well. Please briefly mention the number of replicates earlier in this section for transparency.
L122: Define “RST1–RST5” upon first mention for clarity.
L225-226: Clarify the normalization method used in TMT ratio calculation. Clarify if the TMT ratios were log2-transformed before statistical testing. This is important for interpreting volcano plots later.
L254: State the number of biological and technical replicates clearly (even if mentioned elsewhere) to support reproducibility.
L309–315: Please consider briefly explaining what the first two principal components represent biologically. 96.5% variance explained by PC1 is unusually high. Consider providing biological interpretation: what does PC1 represent? Is it ripening stage, damage response, or both?
L330–333: The GO results are summarized well. Including enrichment p-values for key terms in the main text would improve impact.
L421–429: Adding a sentence to interpret biological implications (e.g., how ethylene response correlates with visible fruit softening) would improve connection to phenotype.
L455–462: Please mention one or two specific proteins or functional groups that best exemplify the proteomic shifts observed—this would help ground the reader in the data.
L495–501: The ethylene modulation discussion is excellent. However, could you further clarify whether the reduction in ethylene-related proteins at D12 is consistent with slower ripening or a protective delay in tissue degradation?
L548–562: Add a clearer transition from your current findings to future PTM studies.
All figures: Ensure each figure has a self-contained legend that explains abbreviations.
L589: The link to supplementary materials is nonfunctional (“https://www.mdpi.com/article/doi/s1”). Please update with the correct DOI or file name.
Reviewer 3 Report
Comments and Suggestions for Authors
Dear authors. I have reviewed the manuscript entitled:
Proteomic analysis of mechanical injury effects in papaya fruit at two maturity stages In general the manuscript is well written and organized. Nevertheless, I would like authors present the list of up and down regulated proteins with the ID of Arabidopsis and papaya, for each D2 and D12 treatments, in order to analyze in the string database v12.0 the molecular mechanisms involved in the damage process. I will consider as major revisionAuthor Response
Please see the attachment.

Reviewer 4 Report
Comments and Suggestions for Authors
This study presents a comprehensive proteomic analysis of the molecular responses to mechanical injury in papaya fruit at two ripening stages. The work addresses a significant gap in postharvest biology by elucidating stage-specific proteomic changes, which could inform strategies to mitigate economic losses. The experimental design is robust, combining histological, biochemical, and advanced mass spectrometry techniques. However, several aspects require clarification or improvement to enhance the manuscript’s impact and reproducibility.
1.The hypothesis (mechanical damage alters proteins linked to oxidative stress, degradation, and phenylpropanoid pathways) is appropriately tested. However, the introduction could better emphasize the novelty of comparing two ripening stages, as most prior work focuses on single-timepoint analyses.
2.The phenol-based method is justified for recalcitrant tissues, but the protocol lacks critical details.
3.The rationale for using TMT 6-plex should be explained, given potential ratio compression issues.
4.Reliance on NetGO 2.0 for a non-model species is reasonable, but cross-validation with BLAST or InterProScan would strengthen functional predictions.
5.The DPP threshold of 50 for pathway perturbation seems arbitrary. Justify this cutoff or use established metrics.
6.The biphasic ethylene response (up at D2, down at D12) is intriguing but lacks mechanistic exploration. Discuss potential crosstalk with ABA and its impact on ripening delay.
7.Figure 1: Include individual data points for color measurements to show variability.
8.Figure 5: Cluster labels (c1–c4) are unclear. Provide a legend or descriptive names.
9.“Proteoforms” (Line 29): Define this term early, as it is central to the discussion on PTMs.
10.“Bruising” vs. “mechanical damage”: Use consistently to avoid confusion.
11.State the *n* for biological replicates in figure captions (currently only in Methods).
12.For PCA (Figure 3A), report variance explained by PC3 if >5%.
13.Contrast results with proteomic studies in other climacteric fruits to highlight conserved/divergent responses.
14.Update citations to include recent reviews on fruit wound responses. e.g., Trends in Plant Science, 2023.
Round 2
Reviewer 1 Report
Comments and Suggestions for Authors
Some minor points:
3. What's the proteome coverage and proteome sequence coverage from the current proteomics
workflow in this study? Has RTS (Real time search) MS3 been compared with SPS MS3?
No unfortunately no comparison between both RST MS3 and SPS MS3 was conducted during
this approach. However if Reviewer 1 consider this like a mandatory point to cover we will cover
this comparison.
No it's not a mandatory point.
9. For figure 3, maybe a set of venn diagram to show the overlap and uniqueness between D2/D12 proteins (total, down regulated, up regulated) would be more intuitive.
We appreciate and recognize the idea of using a Venn diagram to illustrate better the results
regarding the D2/D12 intersection. However, we would like to ask if this is mandatory, due to our current timing is ending by the time we reviewed all the comments and opinions.
Not a mandatory figure as well, will be 'nice to have'.
12. Figure 7, I like the idea of proposing a model based on proteomics changes, but is there any
metabolomics data to backup the proposed model in similar study as key pathways are heavily
metabolites involved?
Yes there is data and reports covering the issue but not in Carica papaya. However we would like
to ask to Reviewer 1 if this topic is mandatory, we would like to improve the manuscript however we understand this a question related to the manuscript, and not for the edition of the current version.
The proposed model would be more convincing if comparing with any existed metabolomics data (added as references), and perhaps discussion can be expanded and be more intriguing to readers if data between proteomics and metabolomics don't fully agree.
Reviewer 3 Report
Comments and Suggestions for Authors
I have asked you to present the list of up and down regulated proteins with the ID of Arabidopsis and papaya, for each D2 and D12 treatments, in order to analyze in the string database v12.0 the molecular mechanisms involved in the damage process. I could not find it.
Reviewer 4 Report
Comments and Suggestions for Authors
The author has already made a thorough revision of the reviewers' comments.
Author Response
We would like to thanks and acknowledge Reviewer 4 for the ideas and insights poured through the peer reviewing process, as an opportunity to improve our work and the present manuscript.
Round 3
Reviewer 3 Report
Comments and Suggestions for Authors
Dear authors, thank you very much for providing the list of up and down regulated proteins either in Arabidopsis thaliana and Carica papaya. This list is very important since it gives you the opportunity to search carefully all the molecular mechanisms involved. In the discussion part you did not mention the role of RPS6A in the life span, closely related to physical damage, and maybe could explain better your findings. On the other hand, make the list more precise and legible. One by one, not in several grouped in one cell. The last question: Based on your proteome, what type of physical or chemical advantage you can use to increased the life span of papaya fruits?.
